# communications
## earth & environment

# Anthropogenic aerosol and cryosphere changes drive Earth's strong but transient clear-sky hemispheric albedo asymmetry

Michael S. Diamond [1,2 ✉], Jake J. Gristey [1,2,3], Jennifer E. Kay [1] & Graham Feingold [2]

A striking feature of the Earth system is that the Northern and Southern Hemispheres reflect identical amounts of sunlight. This hemispheric albedo symmetry comprises two asymmetries: The Northern Hemisphere is more reflective in clear skies, whereas the Southern Hemisphere is cloudier. Here we show that the hemispheric reflection contrast from differences in continental coverage is offset by greater reflection from the Antarctic than the Arctic, allowing the net clear-sky asymmetry to be dominated by aerosol. Climate model simulations suggest that historical anthropogenic aerosol emissions drove a large increase in the clear-sky asymmetry that would reverse in future low-emission scenarios. High-emission scenarios also show decreasing asymmetry, instead driven by declines in Northern Hemisphere ice and snow cover. Strong clear-sky hemispheric albedo asymmetry is therefore a transient feature of Earth's climate. If all-sky symmetry is maintained, compensating cloud changes would have uncertain but important implications for Earth's energy balance and hydrological cycle.

[1] Cooperative Institute for Research in Environmental Sciences, University of Colorado, Boulder, CO 80309, USA. [2] NOAA Chemical Sciences Laboratory, Boulder, CO 80305, USA. [3] Laboratory for Atmospheric and Space Physics, University of Colorado, Boulder, CO 80303, USA. ✉email: michael.diamond@noaa.gov

Ever since reliable space-based estimates of Earth's albedo (broadband shortwave reflectivity) became available in the mid-1960s, it has been observed that the Northern and Southern Hemispheres (NH and SH, respectively) reflect the same amount of sunlight within measurement uncertainty[1–4]. This hemispheric albedo symmetry appears to be non-trivial in a statistical sense[5,6], yet at present, there exists no generally accepted physical explanation for how this symmetry is maintained (if indeed it is maintained, which has not been proven). State-of-the-art global climate models do not systematically simulate hemispherically symmetric albedos[3–5,7], and despite initial findings[1], Earth's outgoing longwave radiation does not currently exhibit a similar degree of hemispheric symmetry[3,4]. Prior measurements may have been inaccurate, or the real situation may have changed as the NH warmed faster than the SH over the last several decades[8]. The hemispheric imbalance in longwave radiation is balanced by cross-equatorial heat transport, with a northward oceanic heat transport driven by the Atlantic Meridional Overturning Circulation partially offset by southward atmospheric heat transport associated with the northward location of the mean Intertropical Convergence Zone[4,9–11].

Although the all-sky albedo is symmetrical between the hemispheres, its clear-sky and overcast components are markedly asymmetric, with much greater clear-sky reflection in the NH balanced by more abundant and brighter clouds in the SH, particularly in the mid-latitudes[6,12,13]. Here we focus on the asymmetric clear-sky component of the all-sky symmetry.

One explanation for the greater NH clear-sky reflection has to do with the arrangement of the continents: because land surfaces are brighter than the oceans, and most of the continents are in the NH, the NH clear-sky should be brighter than the SH. This hypothesis is supported by utilizing the spectral dimension of albedo to attribute the changes to Earth system properties[14]: the NH is brighter at near-infrared wavelengths associated with reflection from land surfaces and vegetation, whereas the SH is brighter at the visible wavelengths associated with reflection from clouds[3]. The continental component of the NH clear-sky advantage is primarily determined by plate tectonics and thus should be stable on geological (millions of years) timescales. Smaller modulations in land coverage and brightness due to sea-level and land-use change can occur more rapidly, however.

The continent-based explanation of Earth's clear-sky hemispheric albedo asymmetry is not the full story, however, as the atmospheric component of the NH-SH clear-sky asymmetry is known to be larger than the surface component, consistent with greater NH aerosol concentrations (airborne particulate matter)[3,7]. Here we show that the atmospheric component dominates the surface component of the clear-sky asymmetry both because anthropogenic aerosol enhances atmospheric reflection in the Northern Hemisphere and also because the Antarctic surface is substantially brighter than the Arctic surface, nearly canceling the effect of mid-latitude and tropical continental surface reflection. If anthropogenic aerosol and the cryosphere matter for Earth's clear-sky hemispheric albedo asymmetry as much as the land distribution, the clear-sky asymmetry is more ephemeral than generally recognized. And if clouds adjust to maintain all-sky albedo symmetry in the face of clear-sky albedo asymmetry changes (which is not a given), there would be hard-to-predict but potentially important implications for Earth's energy balance, the hydrological cycle, and atmosphere-ocean circulations.

## Results
### Atmosphere and cryosphere controls on Earth's clear-sky hemispheric albedo asymmetry.
To assess the atmospheric and surface contributions to Earth's observed clear-sky albedo, we separate these components in the Clouds and the Earth's Radiant Energy System (CERES) Energy Balanced and Filled product[15,16] using a simple single-layer model of shortwave radiative transfer[17–19] (see Methods). Global maps of total clear-sky reflected shortwave radiation ($R_{clr}$) and its atmospheric and surface contributions are shown in Supplementary Figs. 1, 2a, and 3a. Ocean surfaces are extremely dark, whereas land is generally brighter, with ice and desert surfaces, in particular, reflecting very large quantities of sunlight. Reflection from the atmosphere is more globally uniform, although clear maxima are evident in areas of high aerosol concentration (e.g., East Asia and Sahara outflow). Regions of high topography (e.g., the Andes, Tibet, and Antarctica) have minima in atmospheric reflection due to the simple mechanics of there being a thinner overlying atmosphere than in regions closer to sea-level.

Figure 1 shows this clear-sky decomposition averaged over each hemisphere, as the average NH minus SH difference ($\Delta R_{clr}$), and as the NH-SH difference zonally. In each hemisphere, the atmosphere contributes ~60% of the total clear-sky reflection and the surface ~40%. However, the atmosphere contributes ~80% of the hemispheric contrast, with the surface only contributing 20%, in line with previous findings using a similar decomposition method[3,7].

If the continents are mostly located in the NH, and are brighter than the ocean, what can account for such a small surface contribution to the clear-sky hemispheric albedo asymmetry? The answer is compensation by the cryosphere, specifically by Antarctica. As can be seen from the markers in Fig. 1b and the zonal surface contribution in Fig. 1c, the NH continental advantage is substantial from the tropics to the mid-latitudes. At around 60°, however, the situation reverses dramatically, with the Antarctic reflecting so much more sunlight than the Arctic that a large portion (~4 W m$^{-2}$) of the tropical and mid-latitude continent-based advantage (~5.5 W m$^{-2}$) is erased. (Tropical, mid-latitude, and polar values are defined here as a reflection in each region divided by the full hemispheric area, so the sum of all three regions equals the hemispheric value.) In contrast, the NH atmosphere is more reflective at all latitudes, with peaks in the tropics and poles (related in part to the mechanical effect of Antarctica's high topography).

To better understand the atmospheric component of the clear-sky reflection asymmetry, we analyze total aerosol optical depth (AOD or $\tau_a$) and its contributions from black carbon (BC), dust, organic carbon (OC), sea salt, and sulfate (SO$_4$) aerosols in the Modern-Era Retrospective analysis for Research and Applications, Version 2 (MERRA-2) product[20–22] (Fig. 1d; see Methods). It should be noted that absorbing aerosols like black carbon and dust increases atmospheric reflection less than would fully scattering particles, and because they efficiently reduce transmissivity, will further reduce surface reflection as well. Our results are therefore partially sensitive to aerosol type, although this effect is not expected to be large enough to materially affect any conclusions, at least on the hemispheric scale. The effect may be important for regions with particularly large black carbon or dust changes, however.

The NH dominates AOD in the tropics primarily through a large dust contribution. In the mid-latitudes, sulfate pollution in the NH largely balances sea salt in the SH. Without the (largely anthropogenic) sulfate contribution, the SH would presumably dominate mid-latitude AOD. Carbonaceous aerosols (mainly OC) slightly favor the SH in the tropics and the NH closer to the poles. Since dust and sea salt aerosol is largely "natural" in origin, whereas sulfate is largely due to industrial emissions, it is reasonable to expect that the hemispheric AOD contrast ($\Delta\tau_a$)—and therefore atmospheric $\Delta R_{clr}$—may have been substantially milder in the preindustrial climate.

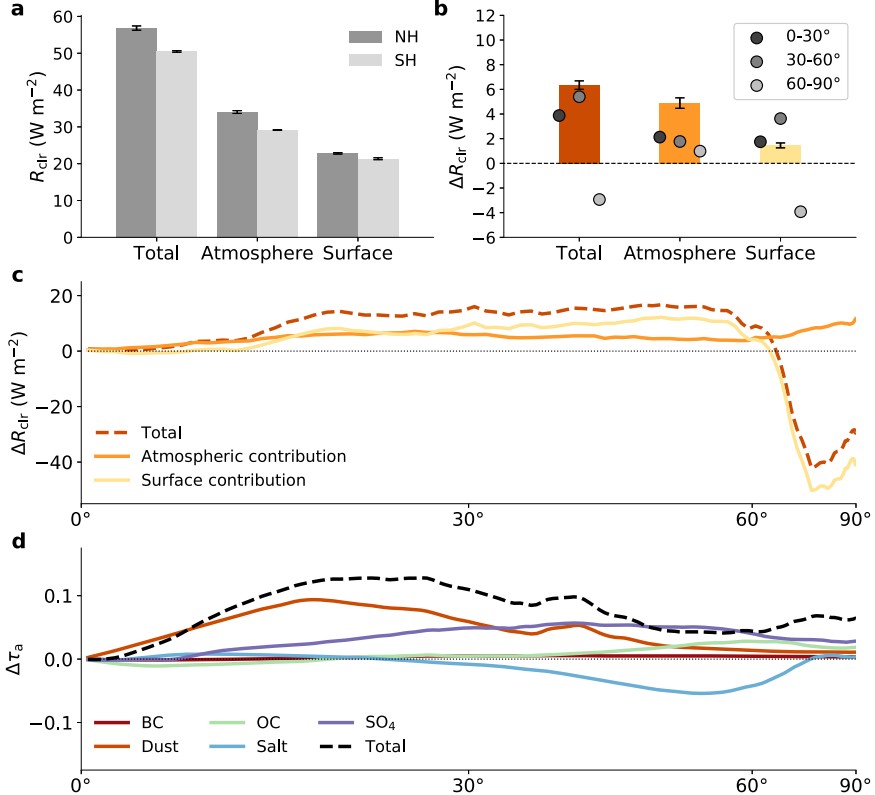

**Fig. 1 Atmospheric and surface contributions to the observed clear-sky hemispheric albedo asymmetry. a** Total clear-sky reflected solar radiation ($R_{clr}$) and its atmospheric and surface components averaged over the Northern Hemisphere (NH; dark gray bars) and Southern Hemisphere (SH; light gray bars). **b** Average NH minus SH differences in $R_{clr}$ ($\Delta R_{clr}$) for the total clear-sky reflection (brown bar) and its atmospheric (orange bar) and surface (yellow bar) components. Markers in **b** indicate the component of the total hemispheric difference attributable to the tropics (0–30°; dark gray), mid-latitudes (30–60°; medium gray), and poles (60–90°; light gray). Error bars in **a**, **b** represent 95% confidence in the mean value. **c** Zonal averages of total $\Delta R_{clr}$ (brown dashed line) and its atmospheric (orange line) and surface (yellow line) components. **d** Zonal hemispheric difference for total AOD ($\Delta \tau_a$; dashed black line) and each species [dark red line for balck carbon (BC), orange line for dust, green line for organic carbon (OC), blue line for sea salt, and purple line for sulfate ($SO_4$)]. For **c**, **d**, the abscissa is area-weighted (plotted as sine of latitude). All averages are for 2003–2020, inclusive.

**Historical evolution of Earth's clear-sky hemispheric albedo asymmetry.** To test this idea, we analyze output from seven coupled climate models that participated in the Aerosol Chemistry Model Intercomparison Project (AerChemMIP)[23] "hist-piAer" experiment, in which aerosol precursor emissions[24] are kept at preindustrial (PI) values, but all else evolves in the same manner as the "historical" experiment (see Methods). Figure 2 shows the clear-sky hemispheric albedo asymmetry and its atmospheric and surface contributions from 1850 to 1865 and 2000–2015 for the historical simulations and from 2000–2015 for the hist-piAer simulations.

For the present-day (PD) period (2000–2015), the models vary by a few W m$^{-2}$ in terms of their total clear-sky asymmetries but diverge more radically from the observations in their breakdown between atmospheric and surface reflection (Fig. 2a–c). No model matches the observed dominance of the atmospheric component over the surface, and some (e.g., MIROC6 and NorESM2-LM) have the ratio reversed as compared to CERES. Supplementary Figs. 2–4 show the difference in $R_{clr}$ (model minus CERES) for the atmosphere ($R_{clr,atm}$) and surface ($R_{clr,sfc}$) globally and as averaged over tropical, mid-latitude, and polar latitude bands, respectively. The underestimates in atmospheric $R_{clr}$ (Supplementary Fig. 2) are mainly in the tropics (particularly over South America, Africa, and Arabia) and NH mid-latitudes (particularly over eastern North America and Europe), and the overestimates in surface $R_{clr}$ (Supplementary Fig. 3) are concentrated over the continents. For MIROC6 in particular, dramatic biases in Antarctic sea ice (Supplementary Fig. 3e) help explain its anomalously low SH surface reflectance (Fig. 2c).

These biases notwithstanding, it is apparent that the atmospheric component of the clear-sky albedo symmetry was much lower (~50–100%) in the PI era (Fig. 2e) or with PI aerosol precursors (Fig. 2h) in the models. [This is partially offset in the total asymmetry by declining NH snow and sea ice cover in the historical PD-PI comparisons (Fig. 2d).] Using GISS-E2-1-G as an example (Fig. 2j–l), the time-series of the full historical and hist-piAer simulations show that the divergence in the experiments (toward greater asymmetry in the historical total and atmospheric $R_{clr}$) takes off around the Second World War period (~1935–1950) and is largely complete by the 1960s and 1970s—which happens to be the earliest time period in which we have space-based observations of Earth's albedo. It is thus possible that we only have reliable observations starting from a relatively unusual time for Earth's clear-sky albedo asymmetry.

The model asymmetry in AOD is highly correlated (Pearson's $r = 0.93$) with the asymmetry in the atmospheric component of $R_{clr}$ (Fig. 3), consistent with a leading role of aerosol in driving variability in $R_{clr,atm}$ over space and time. Changes in $\Delta \tau_a$ are driven primarily by increasing sulfate aerosol in the NH (Supplementary Fig. 5). CERES/MERRA-2 values are a high $\Delta R_{clr,atm}$ outlier compared to the model-based regression fit (Fig. 3; see Methods). Relatedly, the fit based on interannual variability in the asymmetries from CERES and MERRA-2 suggests a much steeper increase in $\Delta R_{clr,atm}$ for an increase in $\Delta \tau_a$ than seen in the model ensemble. This is not likely due to the different methods of calculating the fit [interannual variability

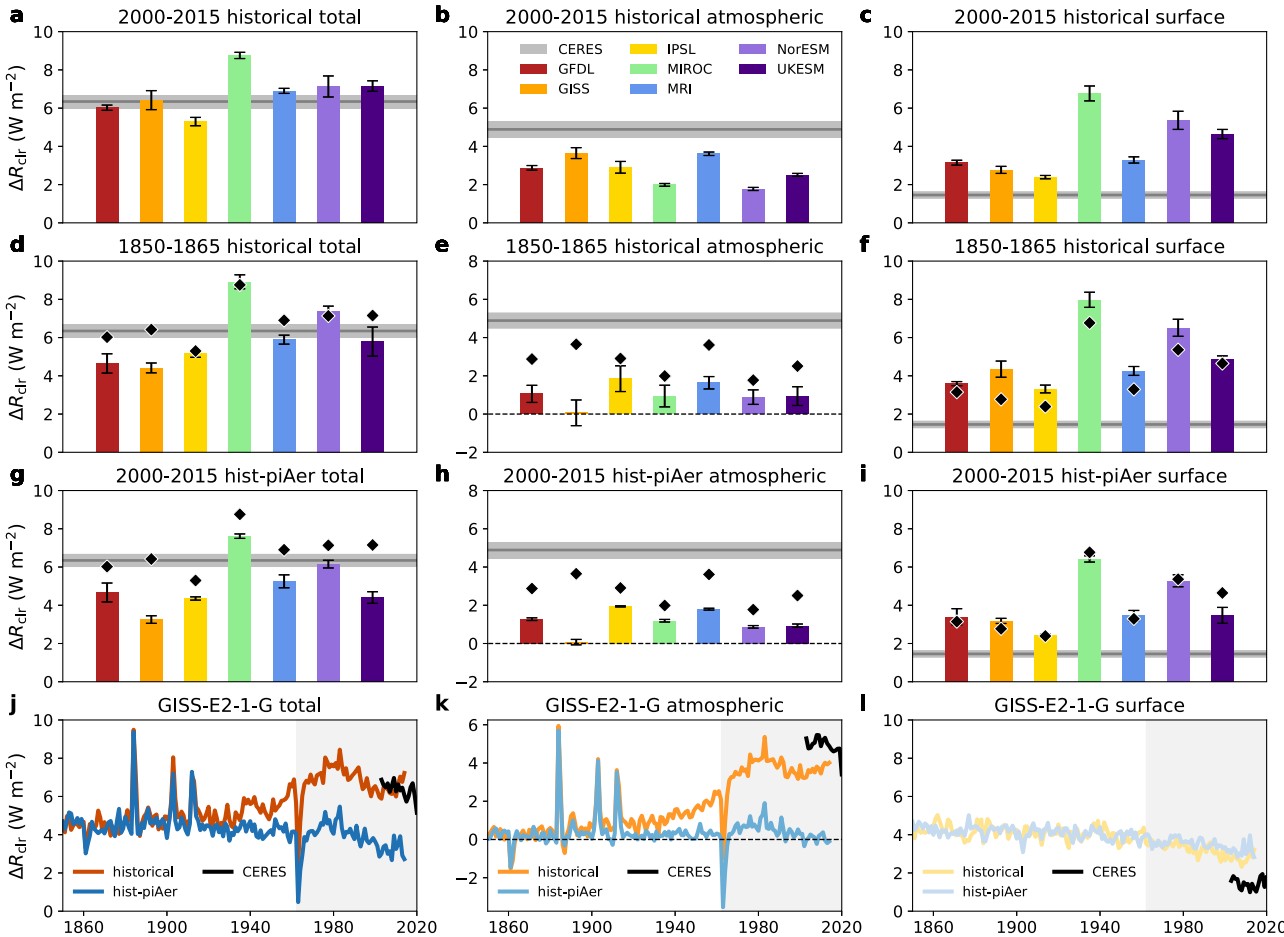

**Fig. 2 Clear-sky hemispheric albedo asymmetry changes in historical CMIP6 runs.** Average of the Northern Hemisphere minus Southern Hemisphere total clear-sky reflection ($\Delta R_{clr}$) and its atmospheric and surface contributions for the 2000–2015 period in the historical runs (**a–c**), 1850–1865 period in the historical runs (**d–f**), and 2000–2015 in the hist-piAer runs (**g–i**). Each model is shown as a colored bar: red for GFDL-ESM4 (GFDL), orange for GISS-E2-1-G (GISS), yellow for IPSL-CM6A-LR (IPSL), green for MIROC6 (MIROC), blue for MRI-ESM2-0 (MRI), indigo for NorESM2-LM (NorESM), and violet for UKESM1-0-LL (UKESM). Error bars for each model represent 95% confidence in the mean value. Diamond markers in **d–i** represent each model's 2000–2015 historical run mean value for reference. CERES mean values are shown as gray lines with shading representing the 95% confidence interval. **j–l** Example time-series of historical (orange lines) and hist-piAer (blue lines) total (**j**), atmospheric (**k**), and surface (**l**) reflection asymmetries from the GISS-E2-1-G model and observed CERES mean values (black lines). Light gray shading in **j–l** represents the time period in which reliable space-based estimates of Earth's albedo have been available.

within an O(10 year) period versus the relationship between different periods]: when comparing slopes calculated using only data from 2000 to 2015 versus for all years 1850–2015 in each model, there are no systematic differences (Supplementary Fig. 6). If we, therefore, assume that the CERES/MERRA-2 interannual slope is representative of PD-PI differences, as it is for the models, we can then estimate the PI value of the clear-sky atmospheric albedo asymmetry for a given PI aerosol asymmetry.

To estimate the PI aerosol asymmetry, we use the good correlation ($r = 0.87$) between the global mean AOD in the PD for a given model and its PD-PI change in $\Delta\tau_a$ (Supplementary Fig. 7) as an emergent constraint. Via Monte Carlo simulation (see Methods), we find that the PI value of $\Delta R_{clr,atm}$ was 2.4 W m$^{-2}$ (95% confidence interval of 0.7 to 3.9 W m$^{-2}$), around half the PD value of $4.9 \pm 0.4$ W m$^{-2}$. This would represent a substantial decrease in the total clear-sky hemispheric albedo asymmetry in the PI compared to that observed today.

**Future evolution of Earth's clear-sky hemispheric albedo asymmetry.** If the past had weaker contrast than the present, what about the future? Under very different future scenarios

within the Shared Socioeconomic Pathways (SSPs)[25,26], there is one consistent outcome: the clear-sky hemispheric albedo asymmetry is projected to decline in the coming century (Fig. 4).

For the low-emission SSP1-2.6 ("sustainability") scenario, the atmospheric component of $R_{clr}$ drives a decline in overall asymmetry (Fig. 4a, b) while the surface plays a more minor role (Fig. 4c), consistent with a decline in co-emitted aerosols and precursor gases. In contrast, the high-emission SSP3-7.0 ("regional rivalry") scenario gets the same overall result (Fig. 4g) but is driven by the surface (Fig. 4i), rather than the atmosphere (Fig. 4h), consistent with maintained high emissions of aerosols and their precursors. Results for the intermediate SSP2-4.5 ("middle of the road") scenario are, fittingly, a blend of those from the two other scenarios (Fig. 4d–f). As illustrated by the UKESM1-0-LL results (Fig. 4j–l), the divergence in the scenarios is apparent by the midcentury.

The surface changes in SSP3-7.0 are largely a story of sea ice (Fig. 5). The hemispheric contrast in the sea ice area (see Methods) has a good correlation ($r = 0.77$) with the hemispheric contrast in the surface component of $R_{clr}$. All models show a decline in Arctic sea ice and NH snow and ice cover on land with

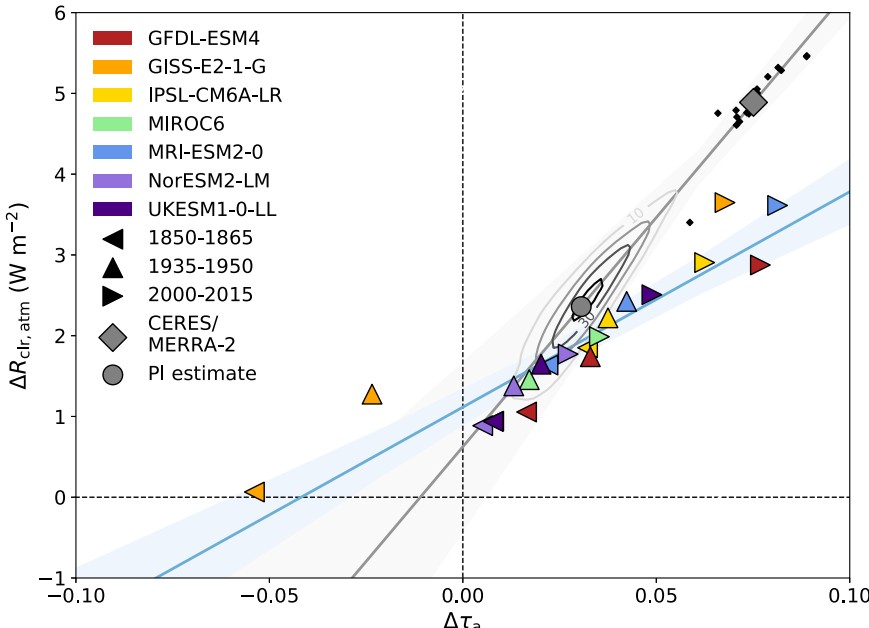

**Fig. 3 Relationship between the hemispheric aerosol and atmospheric reflection asymmetries in the preindustrial, midcentury, and present-day.**
CMIP6 model values of the hemispheric asymmetries in aerosol optical depth ($\Delta\tau_a$) and clear-sky atmospheric reflection ($\Delta R_{clr,atm}$) from the historical runs are represented as colored triangles (facing left for the 1850–1865 mean, up for 1935–1950, right for 2000–2015; red for GFDL-ESM4, orange for GISS-E2-1-G, yellow for IPSL-CM6A-LR, green for MIROC6, blue for MRI-ESM2-0, indigo for NorESM2-LM, and violet for UKESM1-0-LL) and their regression fit and its 95% confidence interval are represented by the blue line and shading. CERES/MERRA-2 data are represented as a gray diamond for the modern mean and as smaller black diamonds for individual years. The regression fit between individual CERES/MERRA-2 years and its 95% confidence interval is represented by the gray line and shading. Contours represent kernel density estimates of the Monte Carlo probabilities (shown every 10 counts) for calculating the preindustrial value of $\Delta R_{clr,atm}$, with the gray circle representing the mean value.

warming, but the magnitude and even sign of Antarctic sea ice changes are more variable (Supplementary Fig. 8). This is in part a mean state issue: models like MIROC6 and NorESM2-LM that have small amounts of Antarctic sea ice in the present-day (Supplementary Fig. 3e, g) have limited room for future declines.

Changes in sea ice albedo[27] and in snow and ice cover on land[28] may also factor into the surface asymmetry changes. For instance, despite nearly equally-balanced trends in NH and SH sea ice area in MRI-ESM2-0 (Fig. 5, Supplementary Fig. 8e, l), there is a modest decline in surface asymmetry associated with decreased reflection over northern and western North America and the Himalayan highlands and Tibetan Plateau.

Due to large internal variability in sea ice concentration[29–31] and outstanding questions about sea ice dynamics, particularly in the Antarctic[32–35], how the surface component of the clear-sky albedo asymmetry would change in reality in a high-warming scenario is subject to more uncertainty than the aerosol-driven atmospheric component. However, it is very plausible that the $R_{clr,sfc}$ asymmetry could substantially decline and even reverse in the future if Arctic sea ice and NH land snow and ice cover are lost more rapidly than Antarctic sea ice in a warming climate.

## Discussion
The changing nature of Earth's clear-sky albedo asymmetry should be observable in the coming decades under any of the future scenarios considered, especially with the planned launch of several visible-shortwave infrared spectroradiometers that will allow for spectral decomposition of reflected sunlight where the signal may be larger[36]. We run simplified radiative transfer simulations (see Methods) to investigate the spectral signal of a decrease in Northern Hemisphere aerosol loading (Supplementary Fig. 9). A cleaner Northern Hemisphere becomes less reflective in the visible spectrum (more associated with reflection from the atmosphere) but has a smaller change in the near-infrared (more associated with reflection from the land surface and vegetation) for all but the highest solar zenith angles. This will exacerbate the present hemispheric differences in the visible and near-infrared reflection[2], which are two portions of the spectrum that will be directly observed separately and with high accuracy during the upcoming Earth radiation budget satellite mission, Libera.

It is tempting to think that we may have glimpsed a lower aerosol and clear-sky asymmetry future during 2020 as a result of the societal response to the COVID-19 pandemic[37,38]. Indeed, 2020 featured low outliers in clear-sky hemispheric albedo asymmetry and aerosol contrast values (Figs. 2j, k, 3, 4j, k). However, the situation is more complicated as the aerosol changes associated with the pandemic lockdowns and economic slowdown were likely too small to be clearly distinguishable above background variability[39–41], and 2020 also featured anomalously large aerosol loadings over the Southern Ocean from the 2019–2020 Australian bushfires[42]. Future work is merited to better understand the unique conditions in 2020 and to what extent they can be used as an "opportunistic experiment" to constrain aerosol radiative forcing[43].

If the observed hemispheric all-sky albedo symmetry is merely the result of chance, these results would remain primarily of academic interest. Indeed, without any compensating cloud-related mechanisms, we should observe an asymmetry in the all-sky reflection in the next few decades and thus obtain a definitively negative answer to the question of whether Earth's hemispheric all-sky albedo symmetry is maintained. However, if clouds respond to the changing clear-sky contrast to maintain all-sky symmetry over the coming decades, there could be important implications for radiative forcing and hydrological and circulation changes depending on the (currently unknown) adjustment mechanism.

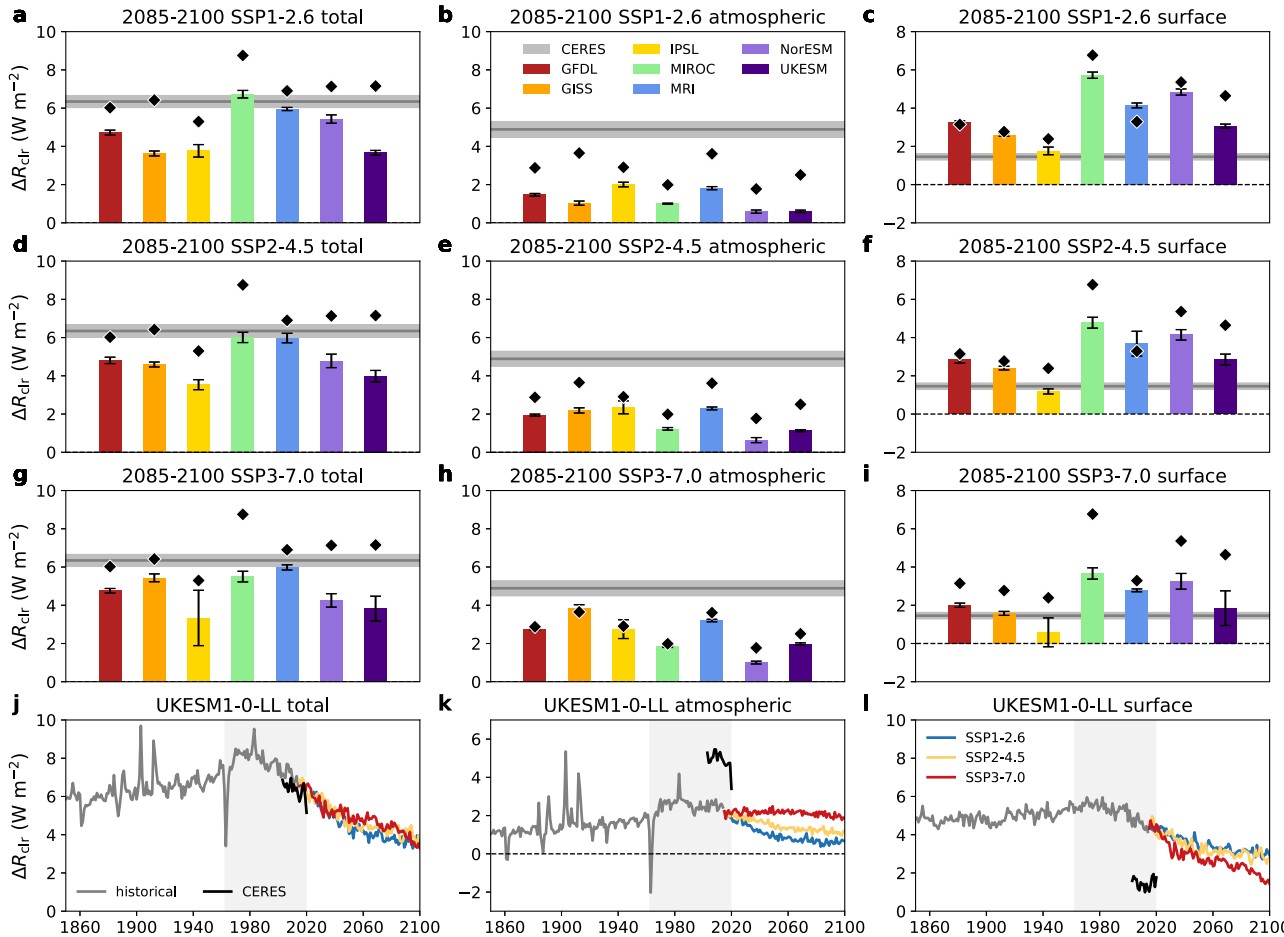

**Fig. 4 Clear-sky hemispheric albedo asymmetry changes in CMIP6 runs of future scenarios. a–i** As in Fig. 2d–i, but for the SSP1-2.6 (**a–c**), SSP2-4.5 (**d–f**), and SSP3-7.0 (**g–i**) runs. **j–l** Example time-series of historical (gray lines), SSP1-2.6 (blue lines), SSP2-4.5 (yellow lines), and SSP3-7.0 (red lines) total (**j**), atmospheric (**k**), and surface (**l**) reflection asymmetries ($\Delta R_{\mathrm{clr}}$) from the UKESM1-0-LL model and observed CERES mean values (black lines). Light gray shading in **j–l** represents the time period in which reliable space-based estimates of Earth's albedo have been available.

If SH clouds darken to compensate for the NH trend, the resulting positive radiative forcing would accelerate global warming. Some worrying evidence that such a hemispheric connection exists comes from CERES reflected shortwave measurements over the past two decades that show nearly equal all-sky darkening trends in the NH and SH[6,44,45]. A strong decline in cloudiness within the northeastern Pacific stratocumulus deck and reductions in aerosol from eastern North America and eastern Asia offer an explanation for the NH trends[45–47], but there is no clear driver for the identical SH trend[6]. Alternatively, if NH clouds brighten to compensate for the clear-sky darkening, global radiative implications could be minor. Of course, the true response (if any) may involve some combination of both NH cloud brightening and SH darkening.

One hypothesized adjustment mechanism involves shifts in the Intertropical Convergence Zone and thus tropical cloudiness toward the darker hemisphere[48]. If this were to occur, it would have important regional implications beyond the global mean precipitation shift[49], particularly for drought-vulnerable locations like the Sahel[50–52]. More recently, attention has shifted to the role played by the extremely cloudy SH mid-latitude oceans[6,7]. Changes in Southern Ocean cloudiness could affect large-scale atmosphere-ocean circulations[13,53–55] and long-term global warming via changes in ocean heat uptake in the Southern Ocean[56–58]. Given the likelihood of large changes in the clear-sky hemispheric albedo asymmetry this century under any plausible

emissions scenario, determining which of these or other as-yet-unidentified mechanisms would likely operate to maintain the all-sky hemispheric albedo symmetry should be a research priority.

## Methods

**Reflected shortwave radiation data**. Clear-sky shortwave fluxes from January 2003 to December 2020 come from the CERES Energy Balanced and Filled Edition 4.1 product and are estimated for the total region (including both cloudy and clear scenes) rather than for only cloud-free portions of scenes using a regional monthly adjustment factor that accounts for the difference between computed fluxes with cloud effects removed and those fluxes when weighted by observed clear-sky fraction[15,16,59]. Clear-sky fluxes estimated in this manner are more comparable with clear-sky output from climate models. Results using clear-sky fluxes from cloud-free portions of scenes only are similar to those shown here.

CERES instruments measure filtered radiances in the shortwave spectrum from 0.3 to 5 μm and fly aboard NASA's polar-orbiting Terra and Aqua satellites as well as the Suomi National Polar-Orbiting Partnership and NOAA-20 satellites[15]. We select data from 2003 to 2020 in which both Terra and Aqua measurements are available. Data from geostationary satellites are used to correct for the full diurnal cycle, and a one-time adjustment (within the range of observational and calibration uncertainty) is applied to ensure that the measured net imbalance in top-of-atmosphere (TOA) radiation matches values from in situ observations of ocean heat uptake[15,60,61]. Surface irradiances are computed independently using the aerosol, cloud, and thermodynamic properties from satellite observations and reanalysis products and are constrained by the TOA irradiances[16].

Uncertainty in the temporal mean values discussed is quantified using the interannual variability assuming a red noise process[62]. Measurement uncertainties are neglected. This approach has the main advantage of allowing us to quantify uncertainty identically between the CERES observations and the CMIP6 models. It is justified because random measurement errors on the order of 1–10 W m⁻² per

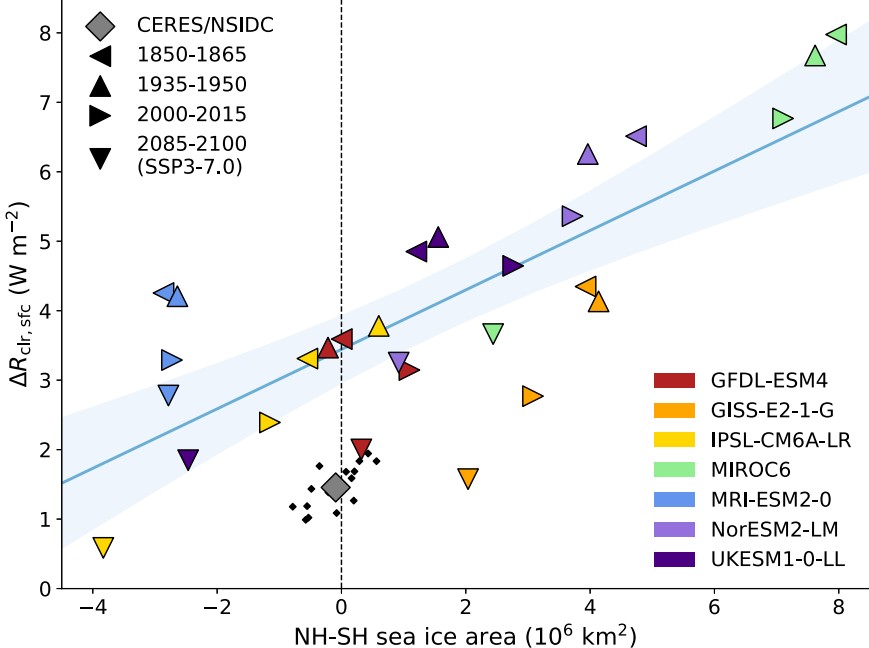

**Fig. 5 Relationship between the hemispheric sea ice and surface reflection asymmetries in the preindustrial, midcentury, present-day, and high-emissions future.** CMIP6 model values of hemispheric asymmetries (NH-SH) in sea ice area and clear-sky atmospheric reflection ($\Delta R_{clr,sfc}$) from the historical and SSP3-7.0 runs are represented as colored triangles (facing left for the 1850–1865 mean, up for 1935–1950, right for 2000–2015, down for 2085–2100; red for GFDL-ESM4, orange for GISS-E2-1-G, yellow for IPSL-CM6A-LR, green for MIROC6, blue for MRI-ESM2-0, indigo for NorESM2-LM, and violet for UKESM1-0-LL) and their regression fit and its 95% confidence interval are represented by the blue line and shading. CERES/National Snow and Ice Data Center (NSIDC) data are represented as a gray diamond for the modern mean and as smaller black diamonds for individual years.

1° × 1° monthly grid box[15,16] rapidly diminish when averaging hemispherically or globally for long time periods [errors of O(0.001–0.01 W m$^{-2}$) as compared to errors of O(0.1–1 W m$^{-2}$) for temporal averaging assuming red noise] and while systematic errors would be more concerning in an absolute sense[17], they would not affect conclusions drawn on the atmosphere/surface breakdown or on hemispheric differences.

Spatiotemporal weighted averaging is performed, accounting for the fact that months have slightly different lengths and that the Earth is oblate, not perfectly spherical. Failure to properly weight by days per month and area can result in errors of O(0.1 W m$^{-2}$) in globally and hemispherically averaged values.

**Aerosol reanalysis data.** Total AOD at 550 nm from MERRA-2 is constrained by assimilation of AOD as retrieved by the Moderate Resolution Imaging Spectro-radiometer instrument aboard the Terra and Aqua satellites, in addition to several other satellite instruments and the Aerosol Robotic Network ground sites, but the breakdown into different species is only constrained indirectly through the total AOD constraint[20]. We, therefore, place greater emphasis on and have greater confidence in the total AOD values than their species decomposition. MERRA-2 does compare well overall with unassimilated satellite and aircraft measurements of aerosol column optical properties and vertical extinction profiles, however, lending some greater confidence[21]. MERRA-2 AOD behaves similarly to other reanalysis products and generally compares well with various observational datasets[63], making it unlikely that the choice to focus on MERRA-2 as opposed to another equally suitable product has any bearing on our results or conclusions. Uncertainty in temporal mean values is quantified assuming red noise[62], as for the reflection data. MERRA-2 data is analyzed from January 2003 to December 2020 to match the CERES record.

**Sea ice concentration data.** Sea ice area data from passive microwave remote sensing observations from January 2003 to December 2020 come from the National Snow and Ice Data Center (NSIDC) Sea Ice Index Version 3 product[64]. Weighting sea ice area by insolation improves its correlation with $R_{clr,sfc}$ for each hemisphere separately but has a negligible impact on the hemispheric difference.

**Climate model data.** Seven state-of-the-art global climate models (abbreviated names in parentheses) from the Coupled Model Intercomparison Project Phase 6 (CMIP6) archive[65] are selected based on their participation in the Aerosol Chemistry Model Intercomparison Project (AerChemMIP) hist-piAer experiment[23] and the Scenario Model Intercomparison Project (ScenarioMIP) SSP1-2.6, SSP2-4.5, and SSP3-7.0 experiments[26]: NOAA Geophysical Fluid Dynamics Laboratories GFDL-ESM4 (GFDL)[66–68]; NASA Goddard Institute for

Space Studies GISS-E2-1-G (GISS)[69–71]; Institut Pierre-Simon Laplace IPSL-CM6A-LR (IPSL)[72–74]; University of Tokyo, National Institute for Environmental Studies, and Japan Agency for Marine-Earth Science and Technology MIROC6 (MIROC)[75–77]; Japan Meteorological Agency Meteorological Research Institute MRI-ESM2-0 (MRI)[78–80]; Norwegian Earth System Model Climate Modeling Consortium NorESM2-LM (NorESM)[81–83]; and the UK Met Office Hadley Centre-Natural Environment Research Council UKESM1-0-LL (UKESM)[84–86].

For models with multiple variants, only one is selected for analysis per model: r1i1p1f1 (GFDL, IPSL, MIROC6, MRI, NorESM); r1i1p3f1 (GISS); and r1i1p1f2 (UKESM).

Temporal averaging accounts for the different calendars used by each model (Gregorian for IPSL, MIROC, and MRI; Gregorian without leap years for GFDL, GISS, and NorESM; and uniform 30-day months for UKESM), and spatial averaging uses atmospheric grid box area for the radiation and aerosol (AOD at 550 nm) fields and either the atmospheric or oceanic grid box area for sea ice depending on the model and its archived output. Not weighting by days per model month can result in errors of O(0.01–0.1 W m$^{-2}$) in globally and hemispherically averaged values. Uncertainty in temporal means is calculated assuming a red noise process, as in the observations.

**Decomposition of top-of-atmosphere reflection into atmospheric and surface components.** Following Donohoe & Battisti[17], we calculate the atmospheric component of TOA planetary albedo using the relation:

$$A = \alpha_{atm} + \alpha_{sfc} \frac{\mathcal{T}^2}{(1 - \alpha_{atm}\alpha_{sfc})}, \qquad (1)$$

where $A$ is the planetary albedo (calculated as the ratio of upwelling to downwelling shortwave radiation at TOA), $\alpha_{atm}$ is the atmospheric component of the planetary albedo, $\alpha_{sfc}$ is the surface albedo (calculated as the ratio of upwelling to downwelling shortwave radiation at the surface), and $\mathcal{T}$ is the atmospheric transmissivity (calculated as the ratio of downwelling radiation at the surface to that at TOA). We then calculate $R_{clr}$ and its atmospheric and surface components ($R_{clr,atm}$ and $R_{clr,sfc}$, respectively) by multiplying by the incoming solar radiation flux, $F_\odot$:

$$R_{clr} = R_{clr,atm} + R_{clr,sfc} = F_\odot \alpha_{atm} + F_\odot \alpha_{sfc} \frac{\mathcal{T}^2}{(1 - \alpha_{atm}\alpha_{sfc})}. \qquad (2)$$

As is clear from Eqs. 1 and 2, the surface and atmospheric components are not independent because the surface component depends on the atmospheric albedo and transmissivity (which accounts for both scattering and absorption). Surface component changes resulting from changes in the atmosphere are much smaller than the original atmospheric changes, however. Thus, while changes in

atmospheric aerosol do affect both the atmospheric and surface components, they primarily affect the atmosphere.

The atmosphere/surface reflection decomposition method is identical between CERES and the CMIP6 models.

**Pre-industrial aerosol contrast estimate**. In order to calculate the PI value of the atmospheric component of the hemispheric asymmetry in $R_{clr}$, we need to estimate the PI value of the hemispheric AOD asymmetry and be able to relate the AOD and $R_{clr,atm}$ asymmetries.

To estimate the PI aerosol contrast, we use the ordinary least squares (OLS) regression between the PD value (defined as the 2000–2015 average) of global mean AOD and the difference between the hemispheric AOD asymmetry in the PD from the PI (defined as the 1850–1865 average) in the seven CMIP6 models as an emergent constraint (Supplementary Fig. 7). All averages are inclusive of the starting year and exclusive of the ending year (e.g., the 2000–2015 average includes all months from January 2000 to December 2014). Using the emergent constraint, we then estimate the real PI aerosol contrast using the PD global mean value from MERRA-2.

We use the OLS regression between $\Delta R_{clr,atm}$ from CERES and $\Delta \tau_a$ from MERRA-2 to estimate the PI value of the atmospheric component of the $R_{clr}$ asymmetry. Based on the similarity between the 2000–2015 and 1850–2015 regressions from the CMIP6 models (Supplementary Fig. 6), the modern CERES-MERRA-2 relationship should be valid for extrapolation back to the PI era.

To quantify uncertainty, we use Monte Carlo simulation to generate 10,000 estimates by randomly drawing from t-distributions of the PD global mean AOD value from MERRA-2 (error calculated assuming red noise), the PD-PI difference in hemispheric AOD contrast (error calculated from the OLS regression uncertainty), the PD hemispheric contrast in AOD from MERRA-2 (error calculated assuming red noise) to calculate the PI hemispheric AOD contrast, and finally the PI value of the $R_{clr,atm}$ asymmetry (error calculated from the OLS regression uncertainty). Kernel density estimation of the Monte Carlo results is used for presentation purposes in Fig. 3.

**Spectral albedo calculations**. Spectrally-resolved radiative transfer output used in Supplementary Fig. 9 is calculated with 1D DIScrete Ordinate Radiative Transfer (DISORT) using the Santa Barbara DISORT Atmospheric Radiative Transfer (SBDART) program[87]. Calculations are performed from 0.2–3.0 µm at 0.005 µm spectral sampling. For simplicity, the NH and SH results are derived from a weighted combination of three calculations that each use standard surface and aerosol properties built into SBDART: (1) snow surface and tropospheric aerosol, (2) ocean surface and oceanic aerosol, and (3) vegetated surface and rural aerosol. The averaging weights for the NH/SH are (1)—10%/10%, (2)—58%/77%, and (3)—32%/13%, respectively[6]. AOD is set to 0.2 for present-day NH, 0.1 for present-day SH, and 0.15 for clean NH. All calculations use the US Standard atmosphere.

## Data availability

CERES data are available from the NASA Langley Research Center (https://ceres.larc.nasa.gov/data/). MERRA-2 data are available from the NASA Goddard Earth Sciences Data and Information Services Center (https://disc.gsfc.nasa.gov/datasets?project=MERRA-2). The Sea Ice Index is available from the NSIDC (https://nsidc.org/data/G02135/versions/3). CMIP6 data are available from the Earth System Grid Federation (ESGF) and were downloaded from the US Department of Energy/Lawrence Livermore National Laboratory node (https://esgf-node.llnl.gov/projects/cmip6/). Processed data to aid in recreating the analyses presented here are available from the NOAA Chemical Sciences Laboratory's Clouds, Aerosol, & Climate program (https://csl.noaa.gov/groups/csl9/datasets/).

## Code availability

All python libraries used in the analysis (cartopy[88] with Natural Earth raster and vector map data, matplotlib[89], numpy[90], scipy[91], and xarray[92]) are freely available. Map projections utilized with cartopy include the orthographic and Equal Earth[93] map projections. The SBDART code is available from Paul Ricchiazzi (https://github.com/paulricchiazzi/SBDART).

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

## Acknowledgements
M.S.D. acknowledges funding from the CIRES Visiting Fellows Program through the NOAA Cooperative Agreement with CIRES (NA17OAR4320101). J.J.G. also acknowledges funding from the NOAA Cooperative Agreement with CIRES and additionally acknowledges the Libera project under NASA Contract 80LARC20D0006. Both J.J.G. and G.F. acknowledge funding from the NOAA Atmospheric Science for Renewable Energy (ASRE) program. G.F. additionally acknowledges funding from the NOAA Climate Program Office Earth's Radiation Budget initiative (#03-01-07-001). J.E.K. acknowledges funding from NSF CAREER (1554659). We acknowledge the World Climate Research Programme, which, through its Working Group on Coupled Modelling, coordinated and promoted CMIP6. We thank the climate modeling groups for producing and making available their model output, the ESGF for archiving the data and providing access, and the multiple funding agencies who support CMIP6 and ESGF. We additionally thank the CERES science team for their efforts in producing and making available their radiation data products. We thank Aiden Jönsson, Mike MacFerrin, Priyam Raghuraman, Tim Smith, and Fangfang Yao for helpful comments.

## Author contributions
M.S.D. conceived and designed the study with input from J.J.G., J.E.K., and G.F. M.S.D. performed all analyses except for the radiative transfer calculations, which were performed by J.J.G. M.S.D. wrote the manuscript with input and editing from all coauthors.

## Competing interests
The authors declare no competing interests.
