## [Peer Review File · Communications Earth & Environment]

Web links to the author's journal account have been redacted from the decision letters as indicated to maintain confidentiality.

28th Jun 22

Dear Dr Diamond,

Your manuscript titled "On the rise and fall of Earth's strong clear-sky hemispheric albedo asymmetry" has now been seen by 3 reviewers, and I include their comments at the end of this message. They find your work of interest, but some important points are raised. We are interested in the possibility of publishing your study in *Communications Earth & Environment*, but would like to consider your responses to these concerns and assess a revised manuscript before we make a final decision on publication.

We therefore invite you to revise and resubmit your manuscript, along with a point-by-point response that takes into account the points raised. In particular, for publication in *Communications Earth & Environment* to be appropriate, we require that you tone down or remove unsupported claims regarding novelty and prevailing misconceptions in the community, as suggested by Reviewers # 1 and 3. In addition, we ask that you address the concern raised by Reviewer # 2 regarding the influence of aerosol type. Please highlight all changes in the manuscript text file.

Please use the following link to submit your revised manuscript, point-by-point response to the referees' comments (which should be in a separate document to any cover letter) and the completed checklist:

[link redacted]

We hope to receive your revised paper within six weeks; please let us know if you aren't able to submit it within this time so that we can discuss how best to proceed. If we don't hear from you, and the revision process takes significantly longer, we may close your file. In this event, we will still be happy to reconsider your paper at a later date, as long as nothing similar has been accepted for publication at *Communications Earth & Environment* or published elsewhere in the meantime.

We understand that due to the current global situation, the time required for revision may be longer than usual. We would appreciate it if you could keep us informed about an estimated timescale for resubmission, to facilitate our planning. Of course, if you are unable to estimate, we are happy to accommodate necessary extensions nevertheless.

Please do not hesitate to contact me if you have any questions or would like to discuss these revisions further. We look forward to seeing the revised manuscript and thank you for the opportunity to review your work.

Best regards,

Clare

Clare Davis, PhD
Senior Editor
Communications Earth & Environment

www.nature.com/commsenv/
@CommsEarth

EDITORIAL POLICIES AND FORMATTING

Editorial Policy: [Policy requirements](https://www.nature.com/documents/nr-editorial-policy-checklist.zip)

Furthermore, please align your manuscript with our format requirements, which are summarized on the following checklist:

[Communications Earth & Environment formatting checklist](https://www.nature.com/documents/commsj-phys-style-formatting-checklist-article.pdf)

and also in our style and formatting guide [Communications Earth & Environment formatting guide](https://www.nature.com/documents/commsj-phys-style-formatting-guide-accept.pdf) .

*** DATA: Communications Earth & Environment endorses the principles of the Enabling FAIR data project (<http://www.copdess.org/enabling-fair-data-project/>). We ask authors to make the data that support their conclusions available in permanent, publically accessible data repositories. (Please contact the editor if you are unable to make your data available).

All Communications Earth & Environment manuscripts must include a section titled "Data Availability" at the end of the Methods section or main text (if no Methods). More information on this policy, is available at <http://www.nature.com/authors/policies/data/data-availability-statements-data-citations.pdf>.

If a community resource is unavailable, data can be submitted to generalist repositories such as [figshare](https://figshare.com/) or [Dryad Digital Repository](http://datadryad.org/). Please provide a unique identifier for the data (for example a DOI or a permanent URL) in the data availability statement, if possible. If the repository does not provide identifiers, we encourage authors to supply the search terms that will return the data. For data that have been obtained from publically available sources, please provide a URL and the specific data product name in the data availability statement. Data with a DOI should be further cited in the methods reference section.

REVIEWER COMMENTS:

Reviewer #1 (Remarks to the Author):

Review of "On the rise and fall of Earth's strong clear-sky hemispheric albedo asymmetry" by Michael Diamond et al.

The motivation for the study is the present-day hemispheric symmetry in TOA reflected shortwave radiation (i.e., planetary albedo), which is a fascinating yet not well understood feature of Earth's climate. If the symmetry was the result of some intrinsic compensation mechanisms, this would put a profound constraint on Earth climate and would have important implications for past and future regional changes. Somewhat in contrast to previous work, the authors focus on the clear-sky aspect of the problem, i.e., the quite substantial hemispheric asymmetry in the clear-sky albedo without clouds that makes the all-sky symmetry so fascinating.

The paper is well written and the analysis methods appear valid. I have already reviewed the previous version of the paper that was under review at Nature. In the revised version considered here the authors present a long rebuttal letter, but the actual changes to the manuscript in my view are rather small and most of the reviewer's criticism is discussed away (instead of substantial changes to the manuscript). As a result, I remain to feel that the authors are overselling their results and that their work would fit better into a specialized journal, or maybe into a perspective/opinion article. I hope to explain this in the following.

In the previous version I have criticized that the authors have, maybe without wanting to do so, created the idea that previous work has not recognized the leading role of aerosols in setting the clear-sky albedo asymmetry in the present-day climate. In the revised version, the authors now talk of a traditional, most-cited and/or original view that they claim to here find to be insufficient (e.g., L17 and L70 of the tracked-changes manuscript). I find this wording is still trying to create the picture that the authors are here correcting a "wrong" view that in my view simply does not exist. I thus remain unsatisfied with the broader framing of the paper.

The clear-sky albedo diagnostic itself and the finding of its transient nature are interesting, and the combination of satellite observations and climate model analysis (including historic and future warming runs) makes a good case about the (likely) time dependence of the clear-sky asymmetry, although the analysis method is fairly standard and straightforward. The numbers derived here are subject to quite large uncertainties (i.e., the pre-industrial clear-sky asymmetry is estimated to be within $\sim 0-4 \text{ Wm}^{-2}$, which does not seem a strong constraint), but nevertheless these numbers are helpful reference points.

The paper closes with a discussion about the possible meaning of the transient nature of the clear-sky asymmetry. This discussion is necessary speculative as it hinges on the unanswered question of whether the all-sky symmetry is pure chance or an intrinsic feature, and so gives little further insights. This part might actually fit better into a perspective or opinion piece instead of an original research article.

Reviewer #2 (Remarks to the Author):

The authors investigate clear-sky northern hemisphere and southern hemisphere albedo asymmetry using CERES data and CMIP6 outputs. They separate atmosphere and surface contributions using a simple one-layer model. They show that the atmosphere contributes more than surface to the clear-sky hemispherical albedo asymmetry. As a consequence, when aerosol loading in the northern hemisphere decreases, clear-sky albedo asymmetry is reduced. They show that this is the case using two 15-year periods using CMIP6 runs.

The manuscript is well written and understanding results is straightforward. This is in part, as the authors acknowledged in the introduction section, that a larger contribution of the atmosphere to

the clear-sky hemispherical albedo asymmetry has been pointed out in earlier studies. therefore, a smaller clear-sky hemispherical albedo asymmetry as the aerosol loading in the northern hemisphere being reduced is a corollary to the earlier result. Therefore, the results are not entirely novel.

I have three minor comments. The third one can be major.

The authors link this clear-sky hemispherical albedo asymmetry to all-sky albedo symmetry and argue on line 250 that, "without any compensation mechanism, we should observe an asymmetry in all-sky reflection in the next few decades". This is true when clouds do not change. However, because cloud signal is much larger compared to aerosol, all-sky asymmetry can be maintained by a separate mechanism regardless of the size of the clear-sky albedo asymmetry. Earth may have a way to maintain symmetrical hemispherical albedo by manipulating clouds because all-sky energy input to each hemisphere may need to be balanced.

Because the way of authors compute surface and atmosphere contributions, both contributions change when aerosol loading changes. Two contributions are not entirely independent, although the change in the surface contribution is at least one order of magnitude smaller than the change in atmospheric contribution. This can be seen by taking a derivative of Eq. (1) with respect to α_{atm} . The authors need to recognize this point.

If only the transmission changes, such as by decreasing soot loading without changing α_{atm} (i.e. changing absorption coefficient without changing scattering coefficient), then only surface contribution changes. Because of the way authors compute contributions, the change of the contribution by aerosol change depends on what type of aerosol changes. A smaller clear-sky hemispherical albedo symmetry during the 1850-1865 period depends on the type of aerosols used in CMIP6 runs. I doubt that the sensitivity to aerosol type is large enough to changes conclusion. But at least the authors need to discuss that the contribution is sensitive to aerosol type.

Reviewer #3 (Remarks to the Author):

Review of On the rise and fall of Earth's strong clear-sky hemispheric albedo asymmetry, by Diamond et al.

This is a revised version of the manuscript, that scrutinizes the contributions to interhemispheric asymmetry in clear-sky albedo in satellite observations, and in future projections. It highlights that the current asymmetry is transient, due to the role of cryosphere and atmospheric aerosol in producing it, and discusses implications of potential cloud adjustment to the clear-sky asymmetry.

The topic of interhemispheric albedo symmetry is fascinating, and closer investigation of the underlying clear-sky asymmetry is useful. The findings are interesting, and clearly presented. The investigation of the clear-sky albedo symmetry response under different future scenarios, with varying warming and aerosol loadings is novel and relevant, although the discussion of possible cloud adjustments remains speculative, since they rest on the question whether clouds will adjust or not. My specific comments regarding the regression in Fig 3, the discussion of the COVID lockdown analogy, and Methods description have been addressed in a satisfying way.

Unfortunately, however, the authors have failed to address the central concern in my previous review: the poorly founded claim that this study corrects a prevailing misconception in the community, that the clear-sky albedo asymmetry is determined by land-ocean distribution. The manuscript actually changed very little in this respect, and reading the new version, I see the same message, that this work claims to be the first to point at other factors than land-ocean distribution determining the clear-sky albedo asymmetry. As before, this is not correct, and the authors need to re-frame their – indeed interesting and relevant – results to align with the published literature on the topic. In the current framing, I don't think their work should be published.

Formulations in the paper have changed from "The traditional explanation..." to "The most cited explanation..." (in the abstract) and "The traditional and most-frequently invoked explanation..." (in the introduction), with references to Vonderhaar and Suomi (1971), Ramanathan (1987), Stephens et al (2015, 2016) and Datsoris and Stevens (2021), but this does not solve the problem.

Neither the paper, nor this review of it is a meta-study of the literature and citation paths on this topic, but a brief search shows that this account is at best incomplete.

-Vonderhaar & Suomi (1971) and Ramanathan (1987) indeed point at surface features being compensated by clouds, and are indeed very well cited (more than 200 and more than 400 citations respectively, although likely not only for pointing at the interhemispheric albedo symmetry).

-Stephens et al (2016) (36 citations) also refers to "...the surface radiative fluxes that are slightly more asymmetrical", but without reference to the mentioned Ramanathan (1987) or Vonderhaar&Suomi (1973) in that context.

-Stephens et al (2015), as the authors agree, discuss and quantify contributions to clear-sky asymmetry arising from differences in aerosol distribution (their section 4). This paper, that arguably is central for bringing the interhemispheric symmetry into discussion in more recent times, has 191 citations.

-In addition, as the authors also agree but still leave out in introducing their topic and motivating their study, both earlier Voigt et al (2013) (39 citations), and later Bender et al (2017) (11 citations), and Jonsson and Bender (2021) (1 citation) show or clearly state (and stating should be enough to count for a general argument of the type "most cited view") that atmospheric contributions from aerosol asymmetry, matters, e.g. by comparing clear-sky ocean-only contributions. It is correct that these papers are not as frequently cited, but they do contradict the view that there is a community misconception to be corrected.

-Datsoris and Stevens (2021) is indeed a recent paper that actually focuses on interhemispheric albedo symmetry that does not mention atmospheric contributions, but only land-sea differences as reason for clear-sky asymmetry. Although it may be prominent, to date it also has only one citation. That Datsoris and Stevens (2021) doesn't mention aerosol, when based on then available literature it should, is more a criticism of that particular paper. This may be warranted, but should be formulated as such.

It would be more fair to say that the original explanation for clear-sky albedo asymmetry is land-sea distribution, referring to the early studies in the 70's and 80's, but that after that there has been work pointing at surface as well as atmospheric aerosol contributions, although there are also examples of recent work that fail to acknowledge aerosol influence on clear-sky asymmetry. The current work would build on the existing literature, to confirm those results that include atmospheric contributions, and offer closer quantification of both aerosol and cryosphere contributions to the clear-sky asymmetry, and discussion of the implications of the transient nature of the clear-sky asymmetry, by studying effects of varying aerosol forcing and warming scenarios.

Here are some concrete suggestions for minimum level changes to be made to acknowledge the state of knowledge in the field:

Abstract: I would suggest to remove the sentences "The most-cited explanation..." and "However, it is the atmosphere..." from the abstract. This would avoid the implication that the community currently has the wrong view, and that problem is now fixed, and rather keep the focus on the novel findings of the paper.

Line 54 Please rephrase, as discussed above. You may well say the original explanation was the distribution of continents, but later work has pointed at clear-sky asymmetries related to aerosol, explicitly or implicitly.

Line 63 Please change this formulation, as it is (following the above required changes) unclear what the "However" is contrasting with

Fig 1 Remove the "not the surface" from the figure headline, as it presumes the presumption that the surface contributes more.

Other minor comments

Line 74: that there “would be hard-to-predict ripple effects across the climate system” is both vague and dramatic. Please either rephrase or make concrete.

Line 182: It seems that less polluted and less icy is also an option, they are not necessarily anticorrelated.

Line 253: there could be important implications

(No additional references)

We thank the three reviewers for a second round of thorough and constructive comments. Below, we provide a point-by-point response and explanation of the changes we made to address their concerns. In particular, we would like to highlight that we have fully removed any implications that prior work did not acknowledge atmospheric aerosol as playing a role in the clear-sky asymmetry and have clarified that our contribution is in looking both at the anthropogenic portion of the aerosol problem and at how the cryosphere offsets much of the continent-based surface contribution, and how these anthropogenic aerosol and cryosphere components change over time. Original reviewer comments are in black and author response is in blue.

Reviewer #1 (Remarks to the Author):

Review of "On the rise and fall of Earth's strong clear-sky hemispheric albedo asymmetry" by Michael Diamond et al.

The motivation for the study is the present-day hemispheric symmetry in TOA reflected shortwave radiation (i.e., planetary albedo), which is a fascinating yet not well understood feature of Earth's climate. If the symmetry was the result of some intrinsic compensation mechanisms, this would put a profound constraint on Earth climate and would have important implications for past and future regional changes. Somewhat in contrast to previous work, the authors focus on the clear-sky aspect of the problem, i.e., the quite substantial hemispheric asymmetry in the clear-sky albedo without clouds that makes the all-sky symmetry so fascinating.

The paper is well written and the analysis methods appear valid. I have already reviewed the previous version of the paper that was under review at Nature. In the revised version considered here the authors present a long rebuttal letter, but the actual changes to the manuscript in my view are rather small and most of the reviewer's criticism is discussed away (instead of substantial changes to the manuscript). As a result, I remain to feel that the authors are overselling their results and that their work would fit better into a specialized journal, or maybe into a perspective/opinion article. I hope to explain this in the following.

In the previous version I have criticized that **the authors have, maybe without wanting to do so, created the idea that previous work has not recognized the leading role of aerosols in setting the clear-sky albedo asymmetry in the present-day climate. In the revised version, the authors now talk of a traditional, most-cited and/or original view that they claim to here find to be insufficient** (e.g., L17 and L70 of the tracked-changes

manuscript). I find this wording is still trying to create the picture that the authors are here correcting a "wrong" view that in my view simply does not exist. I thus remain unsatisfied with the broader framing of the paper.

We have now eliminated these statements and any implications that the prior literature did not acknowledge aerosol as a factor in the clear-sky asymmetry and are clear that our novel contribution is focusing on the anthropogenic component of the aerosol and on the cryosphere offset to the continental effect in the present day and on past and future changes.

The clear-sky albedo diagnostic itself and the finding of its transient nature are interesting, and the combination of satellite observations and climate model analysis (including historic and future warming runs) makes a good case about the (likely) time dependence of the clear-sky asymmetry, although the analysis method is fairly standard and straightforward. The numbers derived here are subject to quite large uncertainties (i.e., the pre-industrial clear-sky asymmetry is estimated to be within $\sim 0-4$ Wm^{-2} , which does not seem a strong constraint), but nevertheless these numbers are helpful reference points.

The paper closes with a discussion about the possible meaning of the transient nature of the clear-sky asymmetry. This discussion is necessary speculative as it hinges on the unanswered question of whether the all-sky symmetry is pure chance or an intrinsic feature, and so gives little further insights. This part might actually fit better into a perspective or opinion piece instead of an original research article.

We believe the two paragraphs in question are helpful for establishing why readers of CEE should care about whether Earth's albedo symmetry is maintained or not. Although our results by their nature (being clear-sky) cannot establish a cloud-related mechanism, they suggest that we will either have a negative answer as to whether such a mechanism exists or will be affected by one within a few decades.

Reviewer #2 (Remarks to the Author):

The authors investigate clear-sky northern hemisphere and southern hemisphere albedo asymmetry using CERES data and CMIP6 outputs. They separate atmosphere and surface contributions using a simple one-layer model. They show that the atmosphere contributes more than surface to the clear-sky hemispherical albedo asymmetry. As a consequence, when aerosol loading in the northern hemisphere

decreases, clear-sky albedo asymmetry is reduced. They show that this is the case using two 15-year periods using CMIP6 runs.

The manuscript is well written and understanding results is straightforward. This is in part, as the authors acknowledged in the introduction section, that a larger contribution of the atmosphere to the clear-sky hemispherical albedo asymmetry has been pointed out in earlier studies. therefore, a smaller clear-sky hemispherical albedo asymmetry as the aerosol loading in the northern hemisphere being reduced is a corollary to the earlier result. Therefore, the results are not entirely novel.

I have three minor comments. The third one can be major.

The authors link this clear-sky hemispherical albedo asymmetry to all-sky albedo symmetry and argue on line 250 that, "without any compensation mechanism, we should observe an asymmetry in all-sky reflection in the next few decades". **This is true when clouds do not change. However, because cloud signal is much larger compared to aerosol, all-sky asymmetry can be maintained by a separate mechanism regardless of the size of the clear-sky albedo asymmetry.** Earth may have a way to maintain symmetrical hemispherical albedo by manipulating clouds because all-sky energy input to each hemisphere may need to be balanced.

We agree with the reviewer and intended to say as such originally. We have rewritten for clarity: "Indeed, without any compensating cloud-related mechanisms, we should observe an asymmetry in all-sky reflection in the next few decades and thus obtain a definitively negative answer to the question of whether Earth's hemispheric all-sky albedo symmetry is maintained. However, if clouds respond to the changing clear-sky contrast to maintain all-sky symmetry over the coming decades, there could be important implications for radiative forcing and hydrological and circulation changes depending on the (currently unknown) adjustment mechanism."

Because the way of authors compute surface and atmosphere contributions, both contributions change when aerosol loading changes. Two contributions are not entirely independent, although the change in the surface contribution is at least one order of magnitude smaller than the change in atmospheric contribution. This can be seen by taking a derivative of Eq. (1) with respect to α_{atm} . The authors need to recognize this point.

Thank you for pointing this out. We have now added a discussion within the Methods: "As is clear from Equations 1 and 2, the surface and atmospheric components are not

independent because the surface component depends on the atmospheric albedo and transmissivity (which accounts for both scattering and absorption). Surface component changes resulting from changes in the atmosphere are much smaller than the original atmospheric changes, however. Thus, while changes in atmospheric aerosol do affect both the atmospheric and surface components, they primarily affect the atmosphere."

If only the transmission changes, such as by decreasing soot loading without changing α_{atm} (i.e. changing absorption coefficient without changing scattering coefficient), then only surface contribution changes. Because of the way authors compute contributions, the change of the contribution by aerosol change depends on what type of aerosol changes. A smaller clear-sky hemispherical albedo symmetry during the 1850-1865 period depends on the type of aerosols used in CMIP6 runs. I doubt that the sensitivity to aerosol type is large enough to change conclusion. But at least the authors need to discuss that the contribution is sensitive to aerosol type.

Thank you for pointing this out. We have now added discussions of this point twice in the results section and a new Extended Data Figure showing the aerosol change by type for the seven AerChemMIP models.

CERES/MERRA-2 section: "It should be noted that absorbing aerosol like dust and black carbon increase atmospheric reflection less than would fully scattering particles, and because they efficiently reduce transmissivity, will further reduce surface reflection as well. Our results are therefore partially sensitive to aerosol type, although this effect is not expected to be large enough to materially affect any conclusions, at least on the hemispheric scale. The effect may be important for regions with particularly large black carbon or dust changes, however."

Pre-industrial section: "Changes in $\Delta\tau_a$ are driven primarily by increasing sulfate aerosol in the NH (Extended Data Fig. 5)."

Reviewer #3 (Remarks to the Author):

Review of On the rise and fall of Earth's strong clear-sky hemispheric albedo asymmetry, by Diamond et al.

This is a revised version of the manuscript, that scrutinizes the contributions to interhemispheric asymmetry in clear-sky albedo in satellite observations, and in future projections. It highlights that the current asymmetry is transient, due to the role of cryosphere and atmospheric aerosol in producing it, and discusses implications of

potential cloud adjustment to the clear-sky asymmetry.

The topic of interhemispheric albedo symmetry is fascinating, and closer investigation of the underlying clear-sky asymmetry is useful. The findings are interesting, and clearly presented. The investigation of the clear-sky albedo symmetry response under different future scenarios, with varying warming and aerosol loadings is novel and relevant, although the discussion of possible cloud adjustments remains speculative, since they rest on the question whether clouds will adjust or not. **My specific comments regarding the regression in Fig 3, the discussion of the COVID lockdown analogy, and Methods description have been addressed in a satisfying way.**

We thank the reviewer again for their useful suggestions with all these points and are pleased that our responses have satisfied their concerns.

Unfortunately, however, the authors have failed to address the central concern in my previous review: the poorly founded claim that this study corrects a prevailing misconception in the community, that the clear-sky albedo asymmetry is determined by land-ocean distribution. The manuscript actually changed very little in this respect, and reading the new version, I see the same message, that **this work claims to be the first to point at other factors than land-ocean distribution determining the clear-sky albedo asymmetry**. As before, this is not correct, and the authors need to re-frame their – indeed interesting and relevant - results to align with the published literature on the topic. In the current framing, I don't think their work should be published.

We did not intend to leave this impression in the revised work and agree with the reviewer that we are definitely not the first to point this out in general. We agree that our phrasing here has become an unnecessary distraction from our main point (the transient nature of the asymmetry) and therefore have decided to avoid discussing anything that could be construed as a "misconception in the community." We have revised the text in line with the useful suggestions provided below.

Formulations in the paper have changed from "The traditional explanation..." to "The most cited explanation..." (in the abstract) and "The traditional and most-frequently invoked explanation..." (in the introduction), with references to Vonderhaar and Suomi (1971), Ramanathan (1987), Stephens et al (2015, 2016) and Datsis and Stevens (2021), but this does not solve the problem.

Neither the paper, nor this review of it is a meta-study of the literature and citation paths on this topic, but a brief search shows that this account is at best incomplete.

-Vonderhaar & Suomi (1971) and Ramanathan (1987) indeed point at surface features being compensated by clouds, and are indeed very well cited (more than 200 and more than 400 citations respectively, although likely not only for pointing at the interhemispheric albedo symmetry).

-Stephens et al (2016) (36 citations) also refers to "...the surface radiative fluxes that are slightly more asymmetrical", but without reference to the mentioned Ramanathan (1987) or Vonderhaar&Suomi (1973) in that context.

-Stephens et al (2015), as the authors agree, discuss and quantify contributions to clear-sky asymmetry arising from differences in aerosol distribution (their section 4). This paper, that arguably is central for bringing the interhemispheric symmetry into discussion in more recent times, has 191 citations.

-In addition, as the authors also agree but still leave out in introducing their topic and motivating their study, both earlier Voigt et al (2013) (39 citations), and later Bender et al (2017) (11 citations), and Jonsson and Bender (2021) (1 citation) show or clearly state (and stating should be enough to count for a general argument of the type "most cited view") that atmospheric contributions from aerosol asymmetry, matters, e.g. by comparing clear-sky ocean-only contributions. It is correct that these papers are not as frequently cited, but they do contradict the view that there is a community misconception to be corrected.

-Datseris and Stevens (2021) is indeed a recent paper that actually focuses on interhemispheric albedo symmetry that does not mention atmospheric contributions, but only land-sea differences as reason for clear-sky asymmetry. Although it may be prominent, to date it also has only one citation. That Datseris and Stevens (2021) doesn't mention aerosol, when based on then available literature it should, is more a criticism of that particular paper. This may be warranted, but should be formulated as such.

It would be more fair to say that the original explanation for clear-sky albedo asymmetry is land-sea distribution, referring to the early studies in the 70's and 80's, but that after that there has been work pointing at surface as well as atmospheric aerosol contributions, although there are also examples of recent work that fail to acknowledge aerosol influence on clear-sky asymmetry. The current work would build on the existing literature, to confirm those results that include atmospheric contributions, and offer closer quantification of both aerosol and cryosphere contributions to the clear-sky asymmetry, and discussion of the implications of the transient nature of the clear-sky asymmetry, by studying effects of varying aerosol forcing and warming scenarios.

Here are some concrete suggestions for minimum level changes to be made to acknowledge the state of knowledge in the field:

Abstract: I would suggest to remove the sentences "The most-cited explanation..." and "However, it is the atmosphere..." from the abstract. This would avoid the implication that the community currently has the wrong view, and that problem is now fixed, and rather keep the focus on the novel findings of the paper.

Changed to: "The clear-sky asymmetry has surface and atmospheric components, with the relatively-bright continents and atmospheric aerosol both disproportionately located in the Northern Hemisphere."

Line 54 Please rephrase, as discussed above. You may well say the original explanation was the distribution of continents, but later work has pointed at clear-sky asymmetries related to aerosol, explicitly or implicitly.

Changed to: "One explanation for the greater NH clear-sky reflection has to do with the arrangement of the continents". We then lay out the plausibility of this hypothesis based on the spectral evidence, as presented in Stephens et al. (2015), without implying that any prior work necessarily focused on this explanation to the exclusion of all other factors.

Line 63 Please change this formulation, as it is (following the above required changes) unclear what the "However" is contrasting with

Changed to: "The continent-based explanation of Earth's clear-sky hemispheric albedo asymmetry is not the full story, however, as the atmospheric component of the NH-SH clear-sky asymmetry is known to be larger than the surface component, consistent with greater NH aerosol concentrations (airborne particulate matter)^{3,7}"

Fig 1 Remove the "not the surface" from the figure headline, as it presumes the presumption that the surface contributes more.

Changed caption to: "Atmospheric and surface contributions to the observed clear-sky hemispheric albedo asymmetry."

Other minor comments

Line 74: that there "would be hard-to-predict ripple effects across the climate system" is both vague and dramatic. Please either rephrase or make concrete.

Rephrased to be more concrete and relate to the discussion: "there would be hard-to-predict but potentially important implications for Earth's energy balance, the hydrological cycle, and atmosphere-ocean circulations"

Line 182: It seems that less polluted and less icy is also an option, they are not necessarily anticorrelated.

Changed to: "...toward a cleaner future?"

Line 253: there could be important implications

Changed.

3rd August 22

Dear Dr Diamond,

Your manuscript titled "On the rise and fall of Earth's strong clear-sky hemispheric albedo asymmetry" has now been seen by our reviewers, whose comments appear below. In light of their advice I am delighted to say that we are happy, in principle, to publish a suitably revised version in Communications Earth & Environment under the open access CC BY license (Creative Commons Attribution v4.0 International License).

We therefore invite you to revise your paper one last time to address any remaining concerns of our reviewers. At the same time we ask that you edit your manuscript to comply with our format requirements and to maximise the accessibility and therefore the impact of your work.

EDITORIAL REQUESTS:

Please review our specific editorial comments and requests regarding your manuscript in the attached "Editorial Requests Table". Please outline your response to each request in the right hand column. Please upload the completed table with your manuscript files.

SUBMISSION INFORMATION:

OPEN ACCESS:

Communications Earth & Environment is a fully open access journal. Articles are made freely accessible on publication under a [CC BY license](http://creativecommons.org/licenses/by/4.0) (Creative Commons Attribution 4.0 International License). This license allows maximum dissemination and re-use of open access materials and is preferred by many research funding bodies.

For further information about article processing charges, open access funding, and advice and support from Nature Research, please visit <https://www.nature.com/commsenv/article-processing-charges>

At acceptance, you will be provided with instructions for completing this CC BY license on behalf of all authors. This grants us the necessary permissions to publish your paper. Additionally, you will be asked to declare that all required third party permissions have been obtained, and to provide billing information in order to pay the article-processing charge (APC).

[link redacted]

Best regards,

Clare

Clare Davis, PhD
Senior Editor
Communications Earth & Environment

www.nature.com/commsenv/
@CommsEarth

REVIEWERS' COMMENTS:

Reviewer #1 (Remarks to the Author):

I have reviewed the two previous versions of the manuscript. I have no principal objections to the paper and the analysis shown, and the focus of the time evolving clear-sky asymmetry is an interesting addition to the literature.

Whether the results are "interesting enough" to the audience of this journal is something I feel less comfortable judging. Having seen how the paper has evolved might also make me biased regarding this point, and so I prefer to not make a judgement.

Reviewer #2 (Remarks to the Author):

The authors revised the manuscript based on reviewers' comments. I do not have any further comments on the revised version.

Reviewer #3 (Remarks to the Author):

Review of On the rise and fall of Earth's strong clear-sky hemispheric albedo asymmetry, by Diamond et al.

In this round of revisions, the authors have addressed also my main concern in the previous versions, where the importance of aerosol contribution to clear-sky asymmetry was presented as a novel finding. My concern was that this claim of advancement was not proper, and previous literature was not rightly acknowledged.

The paper now focuses on quantifying the contributions from anthropogenic aerosol and cryosphere changes to clear-sky albedo asymmetry, and discusses potential implications for clouds and climate in future emission scenarios, which to my knowledge has not been done before. The discussion of cloud compensation to clear-sky asymmetry changes is by necessity hypothetical, but it points at important research directions, and should thereby be of interest to the community.

We thank the three reviewers for their final round of reviews and are pleased that all outstanding concerns have been addressed to their satisfaction.

Reviewer #1 (Remarks to the Author):

I have reviewed the two previous versions of the manuscript. I have no principal objections to the paper and the analysis shown, and the focus of the time evolving clear-sky asymmetry is an interesting addition to the literature.

Whether the results are "interesting enough" to the audience of this journal is something I feel less comfortable judging. Having seen how the paper has evolved might also make me biased regarding this point, and so I prefer to not make a judgement.

Reviewer #2 (Remarks to the Author):

The authors revised the manuscript based on reviewers' comments. I do not have any further comments on the revised version.

Reviewer #3 (Remarks to the Author):

Review of On the rise and fall of Earth's strong clear-sky hemispheric albedo asymmetry, by Diamond et al.

In this round of revisions, the authors have addressed also my main concern in the previous versions, where the importance of aerosol contribution to clear-sky asymmetry was presented as a novel finding. My concern was that this claim of advancement was not proper, and previous literature was not rightly acknowledged.

The paper now focuses on quantifying the contributions from anthropogenic aerosol and cryosphere changes to clear-sky albedo asymmetry, and discusses potential implications for clouds and climate in future emission scenarios, which to my knowledge has not been done before. The discussion of cloud compensation to clear-sky asymmetry changes is by necessity hypothetical, but it points at important research directions, and should thereby be of interest to the community.